Workshop at the 6th Symposium on Advances in Approximate Bayesian Inference (non-archival), 2024 1–22

# Gaussian Stochastic Weight Averaging for Bayesian Low-rank Adaptation of Large Language Models

**Emre Onal**                                    EMRE.ONAL@COLUMBIA.EDU
*Department of Computer Science, Columbia University*

**Klemens Flöge** [*]                            KLEMENS.FLOEGE@HELMHOLTZ-MUNICH.DE
**Emma Caldwell** [*]                            EMMA.CALDWELL@HELMHOLTZ-MUNICH.DE
**Arsen Sheverdin**                              ARSEN.SHEVERDIN@HELMHOLTZ-MUNICH.DE
**Vincent Fortuin**                              VINCENT.FORTUIN@HELMHOLTZ-MUNICH.DE
*Department of Computer Science, Technical University of Munich*
*Helmholtz AI, Munich*

## Abstract

Fine-tuned Large Language Models (LLMs) often suffer from overconfidence and poor calibration, particularly when fine-tuned on small datasets. To address these challenges, we propose a simple combination of Low-Rank Adaptation (LoRA) with Gaussian Stochastic Weight Averaging (SWAG), facilitating approximate Bayesian inference in LLMs. Through extensive testing across several Natural Language Processing (NLP) benchmarks, we demonstrate that our straightforward and computationally efficient approach improves model generalization and calibration competitively with comparable, more sophisticated methods for Bayesian inference in LLMs. We further show that our method exhibits greater robustness against distribution shift, as reflected in its improved performance on out-of-distribution tasks. [‡]

## 1. Introduction

In recent years, LLMs have demonstrated exceptionally strong performance across a wide range of natural language processing tasks (Radford et al., 2019; Touvron et al., 2023a; Brown et al., 2020). Due to the extremely large numbers of parameters in modern foundation models like LLaMA (Touvron et al., 2023b) or GPT (OpenAI, 2023), approximate Bayesian inference has been difficult. Moreover, fine-tuning these models on the full number of weights is inefficient and prohibitively expensive for practitioners without an abundance of computational resources. In response to this problem, recent work has explored adapters for parameter-efficient fine-tuning (PEFT; Ding et al., 2023) of LLMs on downstream tasks. Our research focuses on utilizing these newly developed PEFT techniques for Bayesian subspace inference in these tuning parameters. One of the currently most popular PEFT methods is low-rank adaptation (LoRA; Hu et al., 2022), which fine-tunes a model indirectly by freezing the pre-trained weights and introducing a set of low-rank matrices which are injected into several layers throughout the model. While LoRA can help resolve the issue of inefficiency in fine-tuning, the resulting LLMs still suffer from a significant limitation: they have been shown to be overly confident in their predictions and exhibit poor calibration

---

[*] Equal contribution.

[‡]. Our code is available here: https://github.com/fortuinlab/swag-lora

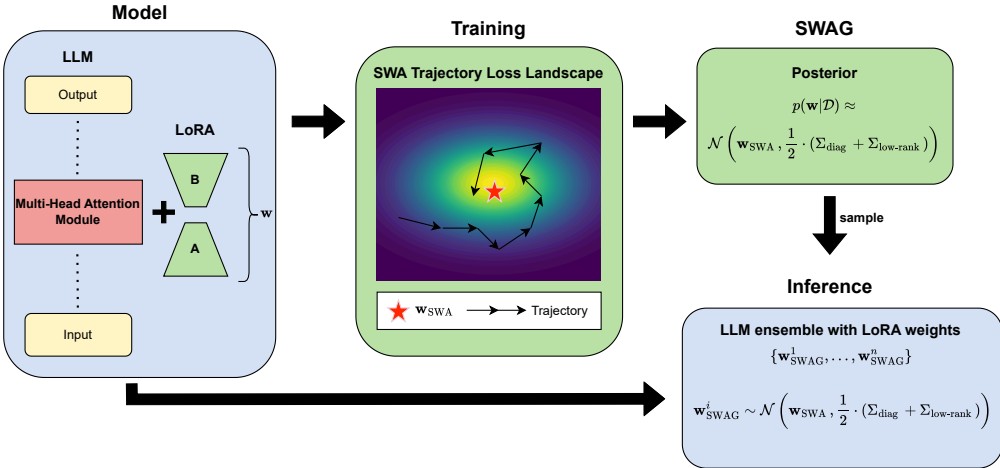

Figure 1: Outline of the SWAG-LoRA training and inference process. The left panel shows the LLM architecture with LoRA fine-tuning (see Section 2.1). The middle and upper right panel depict the SWAG training process, where weight samples are collected across iterations of SGD to calculate the mean and an approximate covariance of the posterior over network weights (see Section 2.2). The lower right panel demonstrates how we form our ensemble of weights for inference by sampling from the learned SWAG posterior.

(Jiang et al., 2021; Xiao et al., 2022; He et al., 2023). The predictive probabilities produced by neural networks in classification settings are often incorrectly interpreted as the confidence of the model. However, models may still be uncertain in their predictions, despite yielding a high softmax output (Gal and Ghahramani, 2016). This phenomenon can be dangerous, especially in safety-critical applications such as medical diagnoses (Singhal et al., 2023), and has prompted the call for better Bayesian treatments of LLMs (Papamarkou et al., 2024).

Many previous works have already proposed various approximate Bayesian inference methods that can be applied in deep neural networks to address this issue (see Appendix A for a detailed treatment of the related work). In this paper, we propose a simple alternative to these methods that avoids any significant numerical and implementation challenges while also consistently improving model generalization and calibration despite its simplicity. We integrate Gaussian Stochastic Weight Averaging (SWAG; Maddox et al., 2019) with LoRA to inexpensively enable approximate Bayesian inference with LLMs.

We evaluate SWA, SWAG, MultiSWA, and MultiSWAG performance against those of baselines such as standard (non-Bayesian) LoRA fine-tuning as well as Monte Carlo (MC) dropout (Gal and Ghahramani, 2016), LoRA ensembles (Wang et al., 2023) and Laplace-LoRA (Yang et al., 2023) on a number of benchmark commonsense reasoning NLU tasks. Most notably, we find that the relatively inexpensive MultiSWAG method achieves competitive performance with the more sophisticated Laplace-LoRA in terms of both accuracy and calibration.

## 2. Methods

### 2.1. Low-Rank Adaptation

Low-Rank Adaptation (LoRA) is a PEFT method that significantly reduces the number of parameters needed for fine-tuning without strongly compromising performance. LoRA introduces low-rank adaptation matrices (A and B in Figure 1) to certain layers (most commonly the query and value projection matrices in attention layers and the classification head (Hu et al., 2022)). It enables efficient and effective fine-tuning with minimal computational resources. Equivalently, it enables the fine-tuning of very large models using limited computational resources. In our context, we leverage LoRA to greatly reduce the number of trainable parameters so that we can feasibly use approximate Bayesian inference methods with LLMs with billions of parameters. LoRA enables us to evaluate our method with LLaMA-2 with 7 billion parameters.

### 2.2. SWA & SWAG

Stochastic weight averaging (SWA) consists of a simple averaging of model weights over the trajectory of Stochastic Gradient Descent (SGD) with data $\mathcal{D}$. Those SGD iterates are obtained using a cyclical or large constant learning rate, which explores the region around the MAP estimate in the loss landscape. SWA has been shown to find flatter solutions than SGD, improving robustness and generalization performance (Izmailov et al., 2018). SWA-Gaussian is a simple extension of SWA, which treats SGD iterates around the MAP as samples from a Gaussian distribution.

Following the proposed method in Maddox et al. (2019), our SWAG implementation uses these SGD iterates to calculate a Gaussian distribution around the SWA estimate. The SWA estimate is the first moment of the SGD iterates, implemented via the running average expressed in Equation (1):

$$\mathbf{w}_{\mathrm{SWA}} = \frac{1}{T} \sum_{i=1}^{T} \mathbf{w}_i \tag{1}$$

The $\mathbf{w}_i$ in Equation (1) refers to the weights of the $i$'th SGD iterate out of a total of $T$ iterates (SGD iterates are generally collected once per epoch).

Our SWAG estimate of the covariance matrix is composed of a diagonal term plus a low-rank approximation of the full covariance matrix. The diagonal term corresponds to a variance estimate for each parameter:

$$\Sigma_{\mathrm{diag}} = \mathrm{diag}(\frac{1}{T} \sum_{i=1}^{T} \mathbf{w}_i^2 - \mathbf{w}_{\mathrm{SWA}}^2) \tag{2}$$

The square operation in Equation (2) is applied elementwise to the parameters of the SGD iterates. This is efficiently implemented using a running average for the second moments of the SGD iterates.

We approximate the sample covariance matrix as shown in Equation (3):

$$\Sigma \simeq \frac{1}{T-1} \sum_{i=1}^{T} (\mathbf{w}_i - \mathbf{w}_{\text{SWA}})(\mathbf{w}_i - \mathbf{w}_{\text{SWA}})^T$$

$$= \frac{1}{T-1} \sum_{i=1}^{T} DD^T \tag{3}$$

In Equation (3), each column of the deviation matrix $D$ represents the i'th SGD iterate's deviation from the sample mean ($\mathbf{w}_{\text{SWA}}$). Since the sample mean $\mathbf{w}_{\text{SWA}}$ is unknown until the end of training, these deviations are instead approximated using the running average sample mean of SGD iterates. Finally, instead of calculating the deviations for each of the $T$ SGD iterates, only the $K$ deviations corresponding to the last $K$ epochs of training are used to calculate a low-rank approximation of the covariance matrix:

$$\Sigma_{\text{low-rank}} = \frac{1}{K-1} \hat{D}\hat{D}^T \tag{4}$$

$\hat{D}$ in Equation (4) refers to the deviation matrix whose columns correspond to the last $K$ deviations $D_i$ for $i = T - K + 1, ..., T$. This low-rank approximation is combined with the diagonal term to obtain the approximate posterior distribution over the weights of the network $\mathbf{w}$:

$$p(\mathbf{w}|\mathcal{D}) \approx \mathcal{N}\left(\mathbf{w}_{\text{SWA}}, \frac{1}{2} \cdot (\Sigma_{\text{diag}} + \Sigma_{\text{low-rank}})\right)$$

The Bayesian Model Average (BMA) can be approximated by sampling from $p(\mathbf{w}|\mathcal{D})$, our posterior distribution over network weights. This approach is particularly attractive for obtaining uncertainties for deep learning models, as it has been shown to often improve generalization, calibration, and uncertainty quantification with minimal computational overhead (Maddox et al., 2019). MultiSWAG is a natural extension of SWAG that leverages an ensemble of SWAG models, enabling an effective mechanism for Bayesian marginalization across multiple modes of the posterior to further improve model generalization and calibration (Wilson and Izmailov, 2020). Similarly, MultiSWA refers to an ensemble of SWA models.

Our implementations of SWAG and MultiSWAG estimate the posterior distribution only over the parameters in the LoRA modules in our network, thereby avoiding the significantly more computationally expensive (particularly memory-intensive) task of estimating a posterior distribution over all network parameters.

## 3. Experiments

### 3.1. Results

Please see Appendix B for a description of our implementation and experimental details, including the models, datasets and calibration and uncertainty quantification metrics we employ.

Table 1: Baseline evaluation of accuracy and calibration of (Multi)SWA(G)-LoRA and other uncertainty-aware LoRA methods with LLaMA-2-7B across benchmark datasets. Our best method, MultiSWAG, consistently outperforms the baselines in both accuracy and calibration, performing on par with more complicated and expensive methods like LLLA and LA. SWA improves accuracy over the baselines at the cost of calibration; Gaussian sampling consistently improves calibration but not accuracy. Results presented are averaged over three separate training runs, with standard errors shown as subscripts. Each best result is bolded, and the second best is underlined.

| Metrics | Methods | OBQA | CQA | ARC-E | ARC-C |
|---------|---------|------|-----|-------|-------|
| ACC ↑ | MAP | $77.9_{0.2}$ | $76.2_{0.3}$ | $78.3_{0.5}$ | $57.8_{0.5}$ |
| | MC Dropout | $77.7_{0.1}$ | $75.8_{0.3}$ | $77.8_{0.7}$ | $55.4_{2.0}$ |
| | Ensemble | $78.0_{0.0}$ | $76.8_{0.2}$ | $77.5_{0.2}$ | $57.6_{0.1}$ |
| | SWA (*ours*) | $81.2_{0.4}$ | $77.0_{0.4}$ | $82.7_{0.2}$ | $63.5_{0.5}$ |
| | MultiSWA (*ours*) | $\mathbf{83.3}_{0.2}$ | $\underline{78.8}_{0.3}$ | $83.7_{0.1}$ | $\mathbf{67.3}_{0.2}$ |
| | SWAG (*ours*) | $81.9_{0.3}$ | $77.0_{0.5}$ | $80.9_{1.1}$ | $62.3_{0.4}$ |
| | MultiSWAG (*ours*) | $\underline{82.8}_{0.5}$ | $\mathbf{79.2}_{0.2}$ | $83.7_{0.1}$ | $\underline{66.5}_{0.0}$ |
| | LLLA (Yang et al., 2023) | $78.7_{0.4}$ | - | $\underline{84.7}_{1.5}$ | $66.2_{0.4}$ |
| | LA (Yang et al., 2023) | $78.9_{0.2}$ | - | $\mathbf{85.1}_{1.5}$ | $65.3_{0.2}$ |
| NLL ↓ | MAP | $0.68_{0.00}$ | $0.69_{0.01}$ | $0.68_{0.00}$ | $1.14_{0.01}$ |
| | MC Dropout | $0.68_{0.01}$ | $0.69_{0.01}$ | $0.68_{0.00}$ | $1.20_{0.07}$ |
| | Ensemble | $0.68_{0.00}$ | $0.65_{0.00}$ | $0.67_{0.01}$ | $1.08_{0.01}$ |
| | SWA (*ours*) | $1.00_{0.03}$ | $0.89_{0.01}$ | $1.06_{0.02}$ | $1.87_{0.03}$ |
| | MultiSWA (*ours*) | $1.00_{0.03}$ | $\underline{0.63}_{0.01}$ | $0.74_{0.01}$ | $1.19_{0.02}$ |
| | SWAG (*ours*) | $\underline{0.60}_{0.02}$ | $0.79_{0.05}$ | $0.62_{0.02}$ | $1.12_{0.04}$ |
| | MultiSWAG (*ours*) | $\mathbf{0.49}_{0.01}$ | $\mathbf{0.59}_{0.01}$ | $\underline{0.53}_{0.00}$ | $\underline{0.91}_{0.01}$ |
| | LLLA (Yang et al., 2023) | $0.97_{0.04}$ | - | $0.87_{0.26}$ | $1.21_{0.16}$ |
| | LA (Yang et al., 2023) | $0.65_{0.01}$ | - | $\mathbf{0.49}_{0.06}$ | $\mathbf{0.88}_{0.03}$ |
| ECE ↓ | MAP | $9.1_{0.2}$ | $6.4_{0.4}$ | $8.3_{0.3}$ | $30.3_{19.4}$ |
| | MC Dropout | $8.7_{0.2}$ | $6.6_{0.1}$ | $7.5_{0.9}$ | $7.8_{1.4}$ |
| | Ensemble | $9.0_{0.3}$ | $\mathbf{4.5}_{0.1}$ | $6.6_{0.0}$ | $\underline{5.3}_{0.7}$ |
| | SWA (*ours*) | $14.9_{0.5}$ | $14.2_{0.4}$ | $14.2_{0.3}$ | $27.9_{0.6}$ |
| | MultiSWA (*ours*) | $8.0_{0.4}$ | $7.7_{0.3}$ | $9.6_{0.3}$ | $14.9_{0.7}$ |
| | SWAG (*ours*) | $\underline{4.8}_{0.4}$ | $11.0_{3.2}$ | $\underline{6.5}_{0.7}$ | $10.9_{1.5}$ |
| | MultiSWAG (*ours*) | $\mathbf{4.7}_{0.2}$ | $\underline{4.7}_{0.1}$ | $6.4_{0.1}$ | $\mathbf{4.9}_{0.5}$ |
| | LLLA (Yang et al., 2023) | $15.8_{0.6}$ | - | $11.6_{2.2}$ | $18.2_{4.4}$ |
| | LA (Yang et al., 2023) | $6.4_{0.8}$ | - | $\mathbf{5.4}_{0.2}$ | $7.4_{0.7}$ |
| Brier ↓ | MAP | $0.33_{0.00}$ | $0.35_{0.00}$ | $0.32_{0.00}$ | $0.58_{0.00}$ |
| | MC Dropout | $0.33_{0.00}$ | $0.35_{0.01}$ | $0.33_{0.01}$ | $0.62_{0.00}$ |
| | Ensemble | $0.33_{0.00}$ | $\underline{0.33}_{0.00}$ | $0.33_{0.01}$ | $0.56_{0.00}$ |
| | SWA (*ours*) | $0.34_{0.01}$ | $0.37_{0.01}$ | $0.31_{0.01}$ | $0.63_{0.01}$ |
| | MultiSWA (*ours*) | $0.26_{0.00}$ | $\mathbf{0.30}_{0.00}$ | $0.26_{0.00}$ | $\underline{0.50}_{0.00}$ |
| | SWAG (*ours*) | $\underline{0.28}_{0.01}$ | $0.36_{0.01}$ | $\underline{0.29}_{0.01}$ | $0.53_{0.01}$ |
| | MultiSWAG (*ours*) | $\mathbf{0.25}_{0.00}$ | $\mathbf{0.30}_{0.00}$ | $\mathbf{0.25}_{0.00}$ | $\mathbf{0.47}_{0.00}$ |

### 3.1.1. Baseline results

In Table 1, we present the test set accuracy (ACC), negative log-likelihood (NLL), expected calibration error (ECE), and Brier score for LLaMA-2-7B fine-tuned on each of our benchmark datasets. We compare (Multi)SWAG against the MAP, MC Dropout, Ensemble, and SWA, as well as the Last Layer Laplace Approximation (LLLA) and full Laplace Approximation (LA) methods proposed in Yang et al. (2023). LLLA and LA approximate the posterior distribution over LoRA parameters in the last layer and all LoRA parameters, respectively. We report the results from their paper in our tables.

We find several interesting results from our experiments. MC Dropout and Ensembles do not appear to significantly affect either the performance or the calibration of the MAP model. This is surprising as other works have demonstrated such improvements from ensembling deep networks (Wang et al., 2023; D'Angelo and Fortuin, 2021).

Our results for ECE are fairly noisy and, as explained in Appendix B, the metric is not as sound as our other calibration metrics. As such, we focus our analyses mainly on the NLL and Brier score to evaluate calibration.

MultiSWAG shows a comparable or higher accuracy to LLLA and LA, while significantly outperforming other methods. SWA and SWAG also exhibit improved accuracy over the baseline methods (MAP, MC dropout, and Ensemble), despite underperforming compared to MultiSWA, MultiSWAG, LLLA, and LA. The Gaussian sampling in SWAG does not appear to further improve accuracy over SWA (nor does it improve accuracy in MultiSWAG over MultiSWA). This implies that MultiSWAG's improvement in generalization performance arises not from the Gaussian sampling but due to a combination of SWA and the exploration of multiple basins of attraction. Neither of these factors alone (SWA or Ensemble) achieves an accuracy similar to MultiSWA or MultiSWAG.

MultiSWAG, LLLA, and LA all achieve comparable NLL, demonstrating consistently better calibration than the baseline methods. MultiSWAG results in the best NLL for OBQA and CQA, and the second best for ARC-E and ARC-C. Comparing SWA to SWAG and MultiSWA to MultiSWAG, we observe that Gaussian sampling significantly improves calibration for a small cost in accuracy. In practice, this tradeoff between calibration and accuracy can be tuned via the sampling scale parameter in SWAG sampling, which controls how much the SWAG samples deviate from the SWA mean. Ensembles and MC dropout do not exhibit improved NLL over the MAP.

Among our methods, MultiSWAG consistently has the lowest Brier score. SWAG obtains the lowest Brier score on three out of four datasets. Note that Yang et al. (2023) did not evaluate the Brier score in their experiments. Hence, the numbers for LA and LA are missing.

### 3.1.2. OOD results

Our results for OOD evaluation and detection are shown in Table 2. Our OOD evaluation results do not display trends as consistently as our baseline evaluation on ID datasets. It appears that SWA, SWAG, MultiSWA, and MultiSWAG consistently have higher OOD accuracy than the MAP, MC Dropout, and Ensemble, but are significantly outperformed by LLLA and LA on ARC-E and ARC-C. Our methods perform relatively similarly to LLLA and LA on MMLU law and MMLU cs. This is interesting, as the distribution shift between

OBQA and ARC-E/ARC-C is smaller than that of our MMLU subsets. Compared to Yang et al. (2023)'s LLLA and LA, our methods' OOD generalization accuracy is significantly worse for near-OOD evaluation but marginally better for far-OOD evaluation.

SWA, MultiSWA, SWAG, and LLLA generally yield worse OOD calibration than the baselines. MultiSWAG, and to a lesser extent LA, appear to have calibration more or less on par with that of the baseline methods. However, there is not sufficient significant evidence to indicate any clear trends between the calibrations of the different methods.

Ensembles demonstrate the best OOD detection performance across all our methods, achieving the highest AUROC for both our uncertainty estimation methods.

## 4. Conclusions

In this work, we demonstrated that the well-known and simple SWAG method can be effectively combined with LoRA to achieve competitive accuracy and calibration, comparable to more sophisticated and computationally expensive techniques recently proposed for Bayesian LoRA in LLMs, such as Laplace-LoRA.

We show our approach's (particularly MultiSWAG-LoRA's) ability to inexpensively improve both model generalization and calibration on a variety of widely used multiple-choice question-answering benchmarks. We also demonstrate that its improved generalization arises from both SWA and the exploration of multiple basins of attraction in the loss landscape, while its improved calibration results from the Gaussian sampling. We also show that our approach is relatively robust to domain shift and can generalize to OOD data better than the MAP.

As language models are incorporated in more real world applications and our dependence on accurate and reliable LLM predictions grows, it becomes ever more important to improve the performance and calibration of these models without introducing prohibitive computational overhead. We hope that this exploration can serve as an incremental step in furthering this important endeavor.

## Acknowledgments

We thank Adam Yang and Laurence Aitchison for helpful discussions. Moreover we want to thank Alexandre Strube and Jan Ebert from the Jülich Supercomputing Centre for their support and advice. VF was supported by a Branco Weiss Fellowship. This work was supported by Helmholtz AI computing resources (HAICORE) of the Helmholtz Association's Initiative and Networking Fund through Helmholtz AI. This work was also supported by the Helmholtz Association Initiative and Networking Fund on the HAICORE@KIT partition.

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

## Appendix A. Related Work

**LLM Fine-tuning** In recent years, a number of methods for making more efficient use of fine-tuning have emerged. Two notable mentions are transfer learning (Houlsby et al., 2019), where a pre-trained LLM is adapted to new tasks or domains, enabling models to leverage vast, pre-learned knowledge bases for a wide range of applications, and zero-shot learning (Wei et al., 2022), where models infer correct responses without prior exposure to specific task examples, which showcases the impressive generalization capabilities of LLMs. While the aforementioned methods use traditional fine-tuning to efficiently generalize and transfer knowledge, Parameter Efficient Fine-Tuning (PEFT; Ding et al., 2022, 2023; He et al., 2023) targets the computational efficiency of the underlying fine-tuning methods.

One particularly noteworthy PEFT method (and the one that we focus on in our experiments) is LoRA (Hu et al., 2022). LoRA introduces a set of low-rank matrices whose outputs are concatenated with several layers throughout the model and optimizes only this comparatively small number of fine-tuning parameters. Consequently, LoRA reduces the number of trainable parameters by three to four orders of magnitude from full model fine-tuning, despite achieving comparable or even superior performance. A key advantage of LoRA is that the low-rank matrices can simply be added to the pre-trained weights, thereby not inducing any increased inference time. As the number of trainable parameters is drastically reduced compared to traditional fine-tuning, the memory requirements of the optimizer states experience the same reduction, which allows for the fine-tuning of much larger models given identical hardware constraints. For instance, the recent study in Chen et al. (2023) has illustrated that fine-tuning LLMs with LoRA can significantly enhance their capacity to handle larger context windows with only a minimal increase in computational costs. Furthermore, LoRA modules trained on different tasks can be stored and used to efficiently switch between models optimized for different downstream tasks, significantly reducing the storage required for the usage of several different fine-tuned LLMs. Recently, a number of other works around LoRA have been published, attempting to further improve its efficiency and flexibility, among others, QLoRA (Dettmers et al., 2023) and GLoRA (Chavan et al., 2023).

**Uncertainty estimation for LLMs** The bulk of previous work on Bayesian inference for LLMs has focused on pre-training (Tran et al., 2019; Xue et al., 2021; Cinquin et al., 2022; Zhang et al., 2020; Chen and Li, 2023), which is quite computationally expensive and additionally does not improve these models much, as large pre-trained models typically already have reasonably good calibration (Kadavath et al., 2022). Additionally, it has been shown that approximate Bayesian inference on posteriors over subspaces of the full parameter space actually produces accurate predictions as well as well-calibrated predictive uncertainty in both regression and classification settings (Izmailov et al., 2020). Therefore, subspace inference is a particularly interesting approach to making LLMs Bayesian.

Recent works that combined language model fine-tuning with ensembles consider either full fine-tuning (Gleave and Irving, 2022; Sun et al., 2022) or introduce ensembles consisting of two members: one trained with full fine-tuning, and the other fine-tuned with LoRA adapters (Hewitt et al., 2021). The first of these methods requires the storage of $M$ sets of model parameters, one for each of the $M$ ensemble members, which can become impractical for the larger LLMs. While the latter method requires fewer parameters to be stored, it is

limited by the small ensemble size. Moreover, Bayesian ideas such as the marginal likelihood have been used for linguistic probing in language models (Immer et al., 2022a).

More recently, methods for combining uncertainties with LoRA were proposed, which are Laplace-LoRA (Yang et al., 2023), LoRA Ensembles (Balabanov and Linander, 2024; Wang et al., 2023), MC-Dropout, and Bayes by Backprop (Andersen and Maalej, 2022). Moreover, non-Bayesian methods of quantifying uncertainties such as conformal prediction (Ye et al., 2024) have also been applied in this context. Laplace-LoRA imposes a Laplace approximation on the posterior over the LoRA parameters, while LoRA ensembles construct an ensemble over the LoRA adapters.

In contrast to these existing works, we show that it is possible to achieve similar results with the use of a simple and inexpensive stochastic weight averaging approach.

**Out-of-distribution (OOD) detection.** Detecting out-of-distribution (OOD) data is crucial for ensuring the reliability and safety of machine learning systems. In open-world scenarios, many models may encounter test samples that are out-of-distribution (OOD), requiring careful handling. Such distributional shifts may arise from semantic changes (Hendrycks and Gimpel, 2017), where OOD samples belong to different classes, or from covariate shifts (Ben-David et al., 2010), where OOD samples come from a different domain.

The integration of Bayesian components into Large Language Models (LLMs) and the resulting uncertainty estimates naturally equip Bayesian LLMs with the capability to detect out-of-distribution (OOD) samples. Given the diverse landscape of Bayesian Deep Learning (BDL) methods and settings, direct comparisons are challenging. However, a recent review by Seligmann et al. (2023), covering a range of BDL approaches in realistic OOD scenarios, found that fine-tuning just the final layers of pre-trained models with BDL algorithms significantly improves both generalization accuracy and calibration on data with realistic distribution shifts, while only slightly increasing runtime overhead. Moreover, these models are often comparable to or even exceed the performance of specialized OOD generalization methods. This finding is supported by the notable enhancements reported by recent Bayesian LoRA-based methodologies (Yang et al., 2023; Balabanov and Linander, 2024; Wang et al., 2023).

Particularly, the post-hoc implementation of Laplace approximation and LoRA fine-tuning, as demonstrated in Yang et al. (2023), has improved calibration and uncertainty estimation. Our work shares the same intuition, drawing on the premise that Bayesian approaches, as they have employed, are instrumental in improving model calibration. This is particularly relevant for LLMs in fine-tuning contexts, where available data is scarce compared to the pre-training phase, highlighting the efficacy of Bayesian methods in navigating uncertainty with limited datasets.

Drawing on the empirical results and insights of the work by Wang et al. (2023), our work is motivated by the findings that LoRA ensembles significantly enhance both the accuracy and calibration of these models, surpassing the outcomes of basic LoRA fine-tuning and other methods like last-layer fine-tuning or Monte Carlo dropout. This evidence lends strong support to the efficacy of ensembling techniques, to which SWAG presented in this work belongs, in not just boosting model performance but also in fine-tuning calibration. Wang et al. (2023) also explore the realm of regularized LoRA ensembles, echoing the classical notion (Breiman, 2001) that diversity among ensemble components is pivotal for

generalization. Recent works (Lakshminarayanan et al., 2017; Kumar et al., 2022; Fort et al., 2019) suggest that these ensembling principles hold true for deep learning architectures as well.

To evaluate how well our models are calibrated and their effectiveness in OOD detection, we utilize established metrics such as negative log-likelihood (NLL), expected calibration error (ECE), and the Brier score (Guo et al., 2017; Osawa et al., 2019). We also report on entropy and cross-model uncertainty metrics, following the argument by Malinin et al. (2020) that the entropy score alone cannot distinguish between epistemic (related to the methodology or model) and aleatoric (inherent to the data) uncertainties, while cross-model uncertainty like model disagreement offers an estimate of epistemic uncertainty only (i.e., model-based). Further discussions are presented in Appendix B.

**Bayesian neural networks.** Bayesian neural networks offer the potential to combine the expressive capabilities of neural networks with the rigorous statistical properties of Bayesian inference (MacKay, 1992; Neal, 1993). However, inference in these complex models has proven to be a challenging endeavor (Jospin et al., 2022), spawning various techniques for approximate inference with different trade-offs between quality and computational cost.

At the one end of the spectrum, we find Markov Chain Monte Carlo (MCMC) approaches, which provide asymptotically correct solutions. Neal (1993); Neal et al. (2011); Welling and Teh (2011); Garriga-Alonso and Fortuin (2021); Izmailov et al. (2021) have contributed to the exploration of these computationally expensive yet theoretically accurate methods. While these methods are considered the "gold standard" for BNN inference, they are already computationally challenging for medium-sized neural network models, let alone for large-scale LLM applications.

Moving towards cheaper approximations, variational methods come into play, offering a range of complexity levels. Researchers have proposed diverse variational approximations, including work by Graves (2011); Blundell et al. (2015); Louizos and Welling (2016); Khan et al. (2018); Osawa et al. (2019). Ensemble-based methods have also been explored as an alternative avenue. This includes recent work by Lakshminarayanan et al. (2017); Wang et al. (2019); Wilson and Izmailov (2020); Ciosek et al. (2020); He et al. (2020); D'Angelo et al. (2021); D'Angelo and Fortuin (2021).

At the other end of the spectrum, we have inexpensive local approximations such as Laplace inference (Laplace, 1774; MacKay, 1992; Khan et al., 2019; Daxberger et al., 2021), which provides a simple and computationally efficient solution. Arguably the cheapest approximations are provided by stochastic weight averaging (Izmailov et al., 2018; Maddox et al., 2019) and MC dropout (Gal and Ghahramani, 2016; Kingma et al., 2015). We explore SWAG in this work, and show that it works surprisingly well given how cheap it is, often performing on par with more expensive methods.

Beyond the challenges related to approximate inference, recent work has also studied the question of prior choice for Bayesian neural networks (e.g., Fortuin et al., 2021, 2022; Fortuin, 2022; Nabarro et al., 2022; Sharma et al., 2023, and references therein). Additionally, model selection within the Bayesian neural network framework has garnered attention (e.g., Immer et al., 2021, 2022b,a; Rothfuss et al., 2021, 2022; van der Ouderaa and van der Wilk, 2022; Schwöbel et al., 2022).

## Appendix B. Experimental Setup

**Implementation Details** All the experiments reported in this paper are conducted using LLaMA-2-7B (Touvron et al., 2023b) (finetuned in 16-bit and evaluated in 32-bit). We leverage the PEFT library (Mangrulkar et al., 2022) implementation of LoRA to adapt the queries, values, and the causal language modeling head of LLaMA-2-7B, using the LoRA rank $r = 8$, $\alpha = 16$, and a LoRA dropout probability of 0.1.

We adapted the official implementation of SWAG (Zellers et al., 2018) to work seamlessly with PEFT's LoRA adapters. We experimented with many schedulers for SWAG and ultimately used a constant learning rate schedule for which we determined the constant learning rate by dividing the fine-tuning optimizer's maximum learning rate by two. We also implemented MultiSWAG, whose performance we evaluate against ensembles of standard LoRA-finetuned LLaMA-2-7B models for a fairer comparison. All ensemble or MultiSWAG models consist of five individual members in the ensemble. When evaluating (Multi)SWAG and MC Dropout, we always sample 15 models from our approximate posterior distributions. Furthermore, for SWAG sampling, we use a sample scale of 1.0.

**Datasets** We evaluate our models on several well-known multiple-choice question answering (MCQA) benchmarks. We evaluate our methods on Open Book Question Answering (OBQA; Mihaylov et al., 2018), Commonsense QA (CQA; Talmor et al., 2019), AI2 Reasoning Challenge Easy and Challenge (ARC-E & ARC-C; Clark et al., 2018), and Measuring Massive Multitask Language Understanding (MMLU; Hendrycks et al., 2021).

The CQA dataset does not offer test set labels. We therefore randomly split the training data into new train and validation sets and use the original validation set to report test set performance. For the remainder of the paper, when speaking of results on the CQA test set, this is the split we will be referring to.

**Calibration and uncertainty quantification** Another challenge that comes along with integrating uncertainty estimation into neural network models is the measurement of the quality of these uncertainties. One of the most common metrics in this regard is the expected calibration error (ECE), which is used to measure the discrepancy between a model's predicted probabilities and the true frequencies of the outcomes. However, it has been shown that most of the commonly used calibration errors are actually lower bounds on the true error (Gruber and Buettner, 2022); particularly, the ECE lower bound (which suffers from shortcomings such as sensitivity to the size of datasets) is the least tight of all the considered metrics, while the Brier score is the tightest. Furthermore, ECE necessitates an empirical decision to be made over the bin sizes for the metric's calculation. For these reasons, in addition to the standard evaluation metrics negative log-likelihood (NLL) and ECE, we also include the Brier score, which gives a better estimate of the true calibration error.

We also evaluate our methods' calibration across both in-distribution (ID) and out-of-distribution (OOD) tasks. In addition to OOD evaluation, we test our methods' uncertainty quantification performance by evaluating whether our uncertainty metrics can reliably detect OOD samples. We investigated several different uncertainty metrics across our experiments and ultimately found that those most effective at detecting OOD samples were the standard entropy calculated over class probabilities (here referred to as *entropy*), as well as

(only for our ensemble/SWAG/MultiSWAG/MC dropout methods) another entropy-based metric we refer to as *average entropy*, which calculates the average of the per-model entropies over all models in the ensemble.

While entropy is the more common method of uncertainty quantification, it has been argued that it is not maximally informative, as it does not actually disentangle epistemic and aleatoric uncertainty (Malinin et al., 2020). Entropy uses the average prediction in the BMA, meaning it cannot distinguish between a scenario in which all hypotheses confidently disagree (epistemic uncertainty) and a scenario in which all hypotheses agree on being highly uncertain (aleatoric uncertainty). Average entropy, on the other hand, averages over the entropy of each ensemble member, meaning that highly confident ensemble members will contribute to lower average entropy scores, even if the individual ensemble member predictions in fact strongly disagree about which class prediction they are confident in. Thus, average entropy places a greater emphasis on measuring aleatoric uncertainty than does the standard entropy metric.

In Section 3.1.2, we report the performance of our methods on OOD datasets and evaluate their OOD detection performance using entropy and average entropy to calculate the area under the ROC curve (AUROC; Liang et al., 2018) (leveraging our uncertainty estimates to classify samples as ID or OOD for different uncertainty thresholds for the classification).

# Appendix C. OOD evaluation and detection results

Table 2: Comparison of OOD evaluation and detection performance of (Multi)SWA(G)-LoRA with other uncertainty-aware LoRA methods. Our methods outperform baselines in accuracy but are generally outperformed by LLLA and LA in both accuracy and calibration. In comparison to LA and LLLA, our methods' OOD generalization accuracy is significantly worse in the near-OOD datasets (ARC-E and ARC-C) but marginally better in the far-OOD datasets (MMLU law and MMLU cs). SWA and SWAG display consistently worse calibration than the baselines, LLLA, and LA. Ensembles consistently yield the strongest OOD detection performance.

| Metrics | Methods | ARC-E | ARC-C | MMLU law | MMLU cs |
|---|---|---|---|---|---|
| ACC ↑ | MAP | $60.2_{0.2}$ | $50.7_{0.3}$ | $34.4_{0.5}$ | $42.8_{0.7}$ |
| | MC Dropout | $60.1_{0.2}$ | $50.8_{0.3}$ | $34.4_{0.5}$ | $42.6_{0.7}$ |
| | Ensemble | $60.6_{0.1}$ | $50.4_{0.3}$ | $34.0_{0.1}$ | $43.4_{0.2}$ |
| | SWA (*ours*) | $66.5_{0.1}$ | $56.0_{0.1}$ | $40.4_{0.1}$ | $46.2_{2.1}$ |
| | MultiSWA (*ours*) | $68.4_{0.4}$ | $58.1_{0.6}$ | $42.4_{0.1}$ | $\mathbf{47.9}_{1.0}$ |
| | SWAG (*ours*) | $64.6_{0.1}$ | $53.5_{0.1}$ | $40.1_{0.23}$ | $44.1_{0.9}$ |
| | MultiSWAG (*ours*) | $67.4_{0.6}$ | $56.8_{0.9}$ | $\mathbf{42.5}_{0.2}$ | $46.1_{0.4}$ |
| | LLLA (Yang et al., 2023) | $78.1_{0.0}$ | $68.1_{0.0}$ | $37.1_{0.0}$ | $45.6_{0.0}$ |
| | LA (Yang et al., 2023) | $\mathbf{78.5}_{0.0}$ | $\mathbf{69.2}_{0.0}$ | $37.3_{0.0}$ | $45.1_{0.0}$ |
| NLL ↓ | MAP | $0.96_{0.00}$ | $1.15_{0.01}$ | $\mathbf{1.40}_{0.01}$ | $\mathbf{1.29}_{0.00}$ |
| | MC Dropout | $0.96_{0.00}$ | $1.15_{0.01}$ | $\mathbf{1.40}_{0.01}$ | $\mathbf{1.29}_{0.00}$ |
| | Ensemble | $0.96_{0.00}$ | $1.15_{0.01}$ | $\mathbf{1.40}_{0.00}$ | $\mathbf{1.29}_{0.00}$ |
| | SWA (*ours*) | $1.43_{0.04}$ | $1.84_{0.05}$ | $2.39_{0.24}$ | $2.56_{0.30}$ |
| | MultiSWA (*ours*) | $1.00_{0.01}$ | $1.24_{0.01}$ | $1.71_{0.03}$ | $1.69_{0.03}$ |
| | SWAG (*ours*) | $1.02_{0.03}$ | $1.28_{0.02}$ | $1.64_{0.10}$ | $1.60_{0.07}$ |
| | MultiSWAG (*ours*) | $0.86_{0.01}$ | $1.05_{0.01}$ | $1.42_{0.00}$ | $1.37_{0.01}$ |
| | LLLA (Yang et al., 2023) | $0.99_{0.00}$ | $1.30_{0.00}$ | $2.06_{0.00}$ | $1.80_{0.00}$ |
| | LA (Yang et al., 2023) | $\mathbf{0.70}_{0.00}$ | $\mathbf{0.90}_{0.00}$ | $1.74_{0.00}$ | $1.35_{0.00}$ |
| ECE ↓ | MAP | $4.9_{0.3}$ | $10.9_{0.5}$ | $15.2_{0.9}$ | $\mathbf{11.4}_{0.2}$ |
| | MC Dropout | $5.0_{0.2}$ | $10.8_{0.5}$ | $15.1_{1.0}$ | $11.6_{0.5}$ |
| | Ensemble | $4.9_{0.5}$ | $11.5_{0.2}$ | $15.8_{0.1}$ | $12.4_{0.3}$ |
| | SWA (*ours*) | $19.6_{1.1}$ | $27.8_{1.5}$ | $33.1_{4.1}$ | $35.3_{3.7}$ |
| | MultiSWA (*ours*) | $10.2_{0.4}$ | $16.8_{0.6}$ | $22.3_{1.7}$ | $23.3_{0.9}$ |
| | SWAG (*ours*) | $7.4_{1.3}$ | $15.5_{0.4}$ | $20.1_{3.6}$ | $20.7_{0.4}$ |
| | MultiSWAG (*ours*) | $\mathbf{3.8}_{0.9}$ | $\mathbf{8.2}_{0.4}$ | $\mathbf{13.9}_{0.6}$ | $15.4_{0.8}$ |
| | LLLA (Yang et al., 2023) | $14.8_{0.0}$ | $21.3_{0.0}$ | $33.6_{0.0}$ | $30.3_{0.0}$ |
| | LA (Yang et al., 2023) | $6.2_{0.0}$ | $8.8_{0.0}$ | $24.7_{0.0}$ | $15.8_{0.0}$ |
| Brier ↓ | MAP | $0.51_{0.00}$ | $0.62_{0.00}$ | $0.75_{0.01}$ | $\mathbf{0.69}_{0.00}$ |
| | MC Dropout | $0.51_{0.00}$ | $0.62_{0.00}$ | $0.75_{0.01}$ | $\mathbf{0.69}_{0.00}$ |
| | Ensemble | $0.50_{0.00}$ | $0.62_{0.00}$ | $0.75_{0.00}$ | $\mathbf{0.69}_{0.00}$ |
| | SWA (*ours*) | $0.53_{0.02}$ | $0.69_{0.02}$ | $0.89_{0.04}$ | $0.88_{0.03}$ |
| | MultiSWA (*ours*) | $\mathbf{0.44}_{0.01}$ | $0.58_{0.00}$ | $0.78_{0.01}$ | $0.75_{0.01}$ |
| | SWAG (*ours*) | $0.48_{0.01}$ | $0.62_{0.01}$ | $0.79_{0.03}$ | $0.75_{0.01}$ |
| | MultiSWAG (*ours*) | $\mathbf{0.44}_{0.00}$ | $\mathbf{0.55}_{0.00}$ | $\mathbf{0.73}_{0.00}$ | $\mathbf{0.69}_{0.00}$ |
| AUROC (entropy) ↑ | MAP | $0.78_{0.00}$ | $0.81_{0.00}$ | $0.92_{0.00}$ | $\mathbf{0.89}_{0.00}$ |
| | MC Dropout | $0.78_{0.00}$ | $0.81_{0.00}$ | $0.92_{0.00}$ | $\mathbf{0.89}_{0.00}$ |
| | Ensemble | $\mathbf{0.94}_{0.00}$ | $\mathbf{0.90}_{0.00}$ | $\mathbf{0.97}_{0.00}$ | $0.86_{0.00}$ |
| | SWA (*ours*) | $0.73_{0.02}$ | $0.76_{0.02}$ | $0.84_{0.05}$ | $0.80_{0.08}$ |
| | MultiSWA (*ours*) | $0.92_{0.00}$ | $0.87_{0.00}$ | $0.96_{0.01}$ | $0.78_{0.01}$ |
| | SWAG (*ours*) | $0.72_{0.02}$ | $0.75_{0.01}$ | $0.84_{0.02}$ | $0.81_{0.01}$ |
| | MultiSWAG (*ours*) | $0.92_{0.00}$ | $0.88_{0.00}$ | $0.95_{0.00}$ | $0.78_{0.01}$ |
| AUROC (average entropy) ↑ | MC Dropout | $0.78_{0.00}$ | $0.81_{0.00}$ | $0.92_{0.00}$ | $\mathbf{0.87}_{0.00}$ |
| | Ensemble | $\mathbf{0.94}_{0.00}$ | $\mathbf{0.90}_{0.00}$ | $\mathbf{0.97}_{0.00}$ | $0.86_{0.00}$ |
| | SWAG (*ours*) | $0.75_{0.00}$ | $0.78_{0.01}$ | $0.88_{0.02}$ | $0.85_{0.00}$ |
| | MultiSWAG (*ours*) | $0.93_{0.00}$ | $0.89_{0.01}$ | $0.96_{0.00}$ | $0.81_{0.01}$ |

