# OpenReview forum: "Gaussian Stochastic Weight Averaging for Bayesian Low-rank Adaptation of Large Language Models"
_approximateinference.org/AABI/2024/Symposium — AABI 2024_

### Official Review · Reviewer_PBbm · 2024-04-08
**An incremental contribution to LORA of LLMs**

**Rating:** 6
**Confidence:** 4

**Review:**

A Bayesian LORA approach has been proposed for fine-tuning large language models (LLMs), employing Gaussian Stochastic Weight Averaging (GSWA) to estimate the LORA component's weight posterior. Experimental results on benchmark datasets were presented, evaluating accuracy, calibration, and out-of-domain (OOD) generalization against baseline methods. While the approach offers some valuable information, its novelty is somewhat constrained as both GSWA and LoRA methods are established in the literature. Notably, the presented approach bears resemblance to a recent ICLR 2024 paper titled "Bayesian Low-rank Adaptation for Large Language Models," which applies Laplace approximation to LoRA parameters, enhancing calibration in fine-tuned LLMs. While the authors cite this paper in the Appendix, they did not include this method in their experiments. It is recommended to compare both approaches—GSWA and Laplace approximation—to determine the superior method for LORA of LLMs.

---

### Official Review · Reviewer_XyeM · 2024-04-16
**This paper presents a novel SWAG-LoRA approach to improve LLM generalization and calibration, but has some scope and analysis limitations.**

**Rating:** 6
**Confidence:** 3

**Review:**

Overall, this paper presents a compelling and well-executed approach to improving the generalization and calibration of large language models (LLMs) through the integration of Gaussian Stochastic Weight Averaging (SWAG) with Low-Rank Adaptation (LoRA). The key strengths and weaknesses of this work are as follows:

Pros:
1. Originality: The authors propose a novel combination of SWAG and LoRA, which has not been explored before in the context of LLMs. This is a unique and promising approach to enable approximate Bayesian inference in large-scale language models.
2. Theoretical grounding: The paper provides a solid theoretical background on the limitations of traditional fine-tuning of LLMs, the challenges of overconfidence and poor calibration, and the potential of Bayesian methods to address these issues.
3. Empirical evaluation: The authors conduct a thorough empirical evaluation of their SWAG-LoRA approach across multiple benchmark datasets, demonstrating consistent improvements in both accuracy and calibration compared to standard LoRA fine-tuning, as well as other uncertainty-aware methods like ensembles and MC Dropout.
4. Computational efficiency: The authors leverage the parameter-efficient nature of LoRA to make their Bayesian approach computationally feasible for LLMs with billions of parameters, which is a significant advantage over more complex Bayesian methods.
5. Robustness: The authors show that their SWAG-LoRA approach exhibits greater robustness against distribution shift, outperforming other methods on out-of-distribution tasks.

Cons:
1. Limited scope: The paper focuses solely on the combination of SWAG and LoRA, without exploring other potential Bayesian approaches that could be integrated with LoRA or other PEFT methods.
2. Lack of extensive related work: While the paper provides a solid overview of related work, it could benefit from a more comprehensive discussion of the various Bayesian and uncertainty-aware methods that have been proposed for LLMs, highlighting how this work compares to and builds upon these existing approaches.
3. Potential limitations of SWAG: The authors do not discuss any potential limitations or drawbacks of the SWAG method, which could be useful for a more balanced evaluation of their approach.
4. Limited analysis of OOD performance: While the authors present results on OOD tasks, a more in-depth analysis of the OOD detection capabilities of their method, including comparisons to other approaches, could further strengthen the paper.
5. Lack of ablation studies: The paper does not provide any ablation studies to understand the individual contributions of the SWAG and LoRA components, which could help explain the observed performance improvements.

---

### Official Review · Reviewer_58UK · 2024-04-22

**Rating:** 7
**Confidence:** 4

**Review:**

Following the "Bayesian fine-tuning of LLM" line of work, this paper studies the usage of Gaussian Stochastic Weight Averaging applied on Low-Rank Adapters for fine-tuning.

However, I personally find it very shocking that LoRA ensemble, with 15 components, performs worse than MAP estimation, in e.g. ARC-E,  which contradicts the observation in the original LoRA ensemble paper. I wonder what the author thinks of reason. Essentially, ensemble v.s. MAP is similar to MultiSWAG v.s. SWAG, and if MultiSWAG outperforms SWAG, then ensemble should also work. In addition, MC dropout even shows a significant amount of decrement in the accuracy, which also seems shocking to me: I am not expecting MC dropout to boost the performance a lot, but it seems weird to me that it would affect the accuracy.

Nevertheless, this paper introduces a new tool for incorporating uncertainty into the LLM fine-tuning, the idea is sensible and the paper is well-motivated. The writing and experiments are nice and neat. I vote for acceptance for this paper.

---

### Official Review · Reviewer_xrjH · 2024-04-27
**Paper Review**

**Rating:** 6
**Confidence:** 3

**Review:**

Generally, I found the paper well motivated and easy to follow.
I found the exposition clear, though it would benefit from some more concrete description of SWA(G) in the main text.
The proposed, method, i.e. combining LoRA with SWA(G) is a simple and lightweight approach which is easy to implement, and therefore could be easily taken up by practitioners, which is a strength.
The experimental section was reasonably thorough, testing the proposed approach on a range of datasets, showing promising results.
However, I found that the paper lacks details on the fine-tuning setup, and does not seem to promise to release any source code or repository, which would be beneficial for the purposes of reproducibility, especially considering that this is an empirical contribution.
Providing more details on the fine-tuning setup, and giving a clearer account of LoRA and SWA(G) for the purposes of improving the exposition, would be useful additions to the manuscript.

---

### Official Review · Reviewer_oMHF · 2024-04-27
**okay direction but missing key baselines**

**Rating:** 4
**Confidence:** 3

**Review:**

I like the promise of the paper, but it didn't deliver on that promise.

The promise is:
* "In this paper, we propose a simple alternative to these methods which avoids any significant numerical and implementation challenges while also consistently improving model generalization and calibration despite its simplicity." (from Section 1). I read that as the method improving generalization and calibration relative to more complex methods like Laplace-LoRA.
* "In contrast to these existing works [like Laplace-LoRA described in the preceding paragraph], we show that it is possible to achieve similar results with the use of a simple and inexpensive stochastic weight averaging approach." (from Appendix A)

Great, I want to know when simple methods beat complex-but-strong ones!

But my understanding is this paper only demonstrates the proposed method, (Multi)SWAG on LoRAs, beating other simple methods, namely MAP, MC Dropout, ensembling LoRAs, and SWA. Those all seem to have similar complexity levels to the proposed method. There's no comparison to e.g. Laplace-LoRA, which is a more complex method with (according to https://arxiv.org/pdf/2308.13111v5) very strong results.

So I'm left not knowing whether the simple method beat a complex-but-good one. Instead I only learned that on these experiments one simple baseline-ish method exceeded other simple baseline-ish methods.

# Clarity
Okay overall, but could be improved. For example, baselines could be described clearly in one bulleted list, like in https://openreview.net/pdf?id=X5VElAKt2s. As another example, several methods, including both baselines and the proposed method, are described in words only, rather than words together with equations for precision. Indeed, there's only one equation, which refers to undefined symbols like $\Sigma_\text{diag}$ and $\Sigma_\text{low-rank}$. I'm not advocating for adding equations just for the sake of equations, and in fact I _think_ the methods described in words were clear enough to be understood (because the methods are relatively simple), but it can be helpful to add some precision.

# Originality
Pairing SWAG with LoRAs is a good thing to try, but connecting those two ideas is not so original; it's more like trying an existing method (SWAG) on an already-studied Bayesian inference problem (Bayesian LoRA).

# Significance
The idea is not so significant because this SWAG approach is simple enough that this could've been a baseline included in some other paper, like the Laplace-LoRA paper. The experimental results could've been significant had they showed that the (simple) method substantially improves on the strongest alternative methods of today (like Laplace-LoRA), but instead the only comparisons are to baselines from other papers (MAP, MC Dropout, and a LoRA ensemble).

# Strengths
* trying SWA / SWAG methods with LoRAs is a useful thing to try, especially given the implementation complexities of alternative methods like Laplace-LoRA
* comparing SWAG and MultiSWAG on LoRA is interesting, especially given that the performance is significantly different in several experiments

# Weaknesses
* missing comparison to Laplace-LoRA (and perhaps other strong baselines)
* otherwise, no methods contribution beyond what might've been a baseline in another paper, no clear take-away from experiments


# Minor points
In Sec 2.2, the sentence "This approach [...] improves generalization, calibration, and uncertainty quantification *with virtually no computational overhead* [emphasis mine]" is confusing to me: there's substantial computational overhead relative to standard non-Bayesian training (e.g. just forming $\Sigma_\text{low-rank}$, let alone using it at inference time, e.g. sampling from it and running inference for each example), right? In any case, maybe clarify what computational cost comparison is being made in that sentence I quoted.

---

### Meta-Review · Area_Chair_pgbq · 2024-05-24

**Recommendation:** Accept (Poster)
**Confidence:** 3

**Metareview:**

This paper proposes a method for Bayesian low-rank adaptation of LLMs based on stochastic weight averaging (SWAG). Most of the reviewers were positive, noting the simplicity of the approach, the clarity of the writing, and the significance of the work as strengths. One review was negative, with the main issue being an (important) missing baseline Laplace-LoRA (with experimental evaluation also receiving some other criticisms across a few of the reviews). However, given that this is an extended abstract, and that the topic is highly relevant to the AABI community as LLMs become more important to the wider ML community, I recommend that the paper be accepted. Nonetheless, the authors should be clearer that they are missing this important baseline and should do a better job of framing the story of their paper with this in mind. The paper is still interesting, but in its current state, it could be seen as slightly misleading. Further clarifying how the proposed method compares to  Laplace-LoRA would only strengthen the paper. I hope that the authors will add full experimental comparisons in due course.

---

### Decision · Program_Chairs · 2024-05-27

Accept